# Identification and Empiric Evaluation of New Inhibitors of the Multidrug Transporter P-Glycoprotein (ABCB1)

**DOI:** 10.3390/ijms24065298

**Published:** 2023-03-10

**Authors:** Yasmeen Cheema, Yusra Sajid Kiani, Kenneth J. Linton, Ishrat Jabeen

**Affiliations:** 1School of Interdisciplinary Engineering & Sciences (SINES), National University of Science and Technology, Sector H-12, Islamabad 44000, Pakistan; 2Blizard Institute, Faculty of Medicine and Dentistry, Queen Mary University of London, London E1 2AT, UK

**Keywords:** ABCB1, MDR, P-glycoprotein, multidrug resistance, pharmacophore, drug efflux

## Abstract

The expression of the drug efflux pump ABCB1 correlates negatively with cancer survival, making the transporter an attractive target for therapeutic inhibition. In order to identify new inhibitors of ABCB1, we have exploited the cryo-EM structure of the protein to develop a pharmacophore model derived from the best docked conformations of a structurally diverse range of known inhibitors. The pharmacophore model was used to screen the Chembridge compound library. We identified six new potential inhibitors with distinct chemistry compared to the third-generation inhibitor tariquidar and with favourable lipophilic efficiency (LipE) and lipophilicity (CLogP) characteristics, suggesting oral bioavailability. These were evaluated experimentally for efficacy and potency using a fluorescent drug transport assay in live cells. The half-maximal inhibitory concentrations (IC_50_) of four of the compounds were in the low nanomolar range (1.35 to 26.4 nM). The two most promising compounds were also able to resensitise ABCB1-expressing cells to taxol. This study demonstrates the utility of cryo-electron microscopy structure determination for drug identification and design.

## 1. Introduction

ABCB1 is a promiscuous ATP-binding cassette (ABC) efflux transporter that is prominent in the apical membranes of polarized epithelial cells of the liver, kidney, blood–brain barrier (BBB), and intestines, where the transporter influences the absorption, distribution, and elimination (ADE) of exogenous (and endogenous) cytotoxic compounds [1,2]. When expressed in cancer cells, ABCB1 can prevent the intracellular accumulation of drugs, including frontline chemotherapeutic agents such as doxorubicin, paclitaxel, and vincristine [3,4]. Innate or upregulated expression of ABCB1 has been identified as an important clinical factor in the treatment of several cancers, first reported in leukaemias [5,6] breast cancer [7], small-cell lung cancer [8], (reviewed in Fletcher et al. 2016 [9] and Robey et al. 2018 [10]). Despite this mechanism of multidrug resistance being identified in the 1970s [11] and the ABCB1 cDNA being cloned in the 1980s [12], the phenomenon remains a significant public health issue today [1].

The development of effective and specific ABCB1-inhibitors has long been a goal to overcome multidrug resistance (MDR) [4,13,14]. In this context, a number of first-, second- and third-generation inhibitors of ABCB1 have been identified and tested, but, so far, none have achieved approval [3,15,16]. The first-generation inhibitors, including amiodarone, cyclosporine A and verapamil, exhibited toxicity and low potency. The second-generation inhibitors, including valspodar and dexverapamil, were comparatively potent but most failed due to poor selectivity and poor pharmacokinetic properties. The third generation of inhibitors, including tariquidar, elacridar and zosuquidar, are more potent and efficacious than the first- and second generation-inhibitors, but were linked to various toxicity-related issues [3]. Collectively, the development of clinically useful ABCB1 inhibitors has been beset with a lack of efficacy, selectivity, and poor toxicity profiles [4,13,15].

The advent of structure data for human ABCB1 [17,18,19] is a watershed moment that may allow optimized design of ABCB1 inhibitors to be more potent, less toxic, and clinically efficacious [16,20,21,22]. Herein, a selective dataset of 98 compounds was used to generate a database to build a ligand-based pharmacophore model using an active analogue approach that incorporated the best docked poses of these compounds. The pharmacophore model was then used to screen the Chembridge database [23] of almost a million drug-like molecules. The six best candidates with favourable lipophilic efficiency profile (LipE) and partitioning coefficient profile (CLogP) were identified and then validated experimentally for inhibition of ABCB1 in live cells using a fluorescent drug accumulation assay. Remarkably, this study revealed that four of the six test compounds had the same efficacy as tariquidar to fully inhibit ABCB1 with potency (IC_50_ values) in the low nanomolar range.

## 2. Results

### 2.1. Data Collection and Template Selection

A dataset of 98 compounds from eight different chemical classes were identified from the literature with apparent selective inhibitory activity for ABCB1 over ABCG2 or ABCC1 [24,25,26,27,28]. The half-maximal inhibitory concentration (IC_50_) values of the dataset were estimated between 0.05 to 113 µM. The dataset was first sub-divided into 74 highly active compounds with IC_50_ values between 0.05 and 10 µM, and 24 least active compounds with IC_50_ values between 14 and 113 µM. The top-scoring docked conformation of the highly active compound Q37 (dataset of selective compounds in Appendix A) with an IC_50_ value of 0.05 µM was selected for building the pharmacophore model.

### 2.2. Pharmacophore Model Generation and Validation

The key features of the pharmacophore model for ABCB1-inhibition were identified using the unified scheme of pharmacophore editor. The final model featured three hydrophobic groups, one aromatic and one hydrogen-bond acceptor (Figure 1). The pharmacophoric features were optimized at different radii ranging from 1.7 Å to 2.0 Å. Hydrophobic feature 1 (Hyd1) and Hyd3 were selected with a radius of 1.7 Å, while Hyd2 was adjusted to a radius of 2.0 Å. The aromatic and hydrogen bond acceptor features were optimized at a radius of 1.8 Å. The final ligand-based pharmacophore model of ABCB1 identified the distance between Hyd1 and Hyd2 as 3.42 Å, Hyd1 and Hyd3 as 5.56 Å, and Hyd2 and Hyd3 as 6.05 Å. Furthermore, the distances between the hydrogen bond acceptor (HBA) and Hyd1, Hyd2, and Hyd3 measured 7.74 Å, 8.27 Å, and 4.64 Å, respectively. Additionally, the three hydrophobic groups (Hyd1, Hyd2, and Hyd3) and one HBA feature were separated from the aromatic (Aro) feature by 4.32 Å, 8.56 Å, 5.39 Å, and 6.00 Å, respectively. The mutual pharmacophoric feature distances (in Å) of the pharmacophore model are summarized in Table 1.

The final pharmacophore model was further validated in silico by test screening against the docked conformations of all other 97 inhibitors in the dataset (excluding Q37) with IC_50_ values between 0.09 and 113 µM. For this purpose, a total of 250 conformations of each compound were generated and further packed using MOE.

To estimate the predictive ability of the pharmacophore model, Matthew’s correlation coefficient (MCC) and model accuracy were calculated using Equations (1) and (2) (described in the Methods section). The model depicted 93% of compounds as True Positive (TP), 79% of compounds as True Negative (TN), 20% of compounds as False Positive (FP), and 6% of compounds as False Negative (FN). Overall, the pharmacophore model accuracy was estimated at 0.89 and the MCC value was 0.72, suggesting statistical significance [29,30]. Therefore, the pharmacophore model was used for the virtual screening of the compound library.

### 2.3. Virtual Screening and Hit Identification

To identify potential hits, the publicly available ChemBridge database [23] of 792,047 compounds was screened virtually against various models and filters (Figure 1). Briefly, preprocessing was performed to remove duplicates and fragments with the molecular weight (MW) < 200 Da and >500 Da, reducing the dataset to 788,149 compounds. Fifty conformations of each compound of this curated dataset were generated stochastically, followed by the packing of the database using MOE software. A hERG filter [30] was used to identify 400,754 hERG non-blockers, which were then screened against our pharmacophore model, resulting in a curated dataset of 149,671 compounds. The hERG (human ether-a-go-go-related gene) is a potassium channel that plays a vital role in cardiac repolarization. Therefore, it is important to avoid hERG inhibition in order to minimise the potential for off-target cardiotoxicity. The dataset was further shortlisted and refined using Cytochrome P450 filters [31], thus, a curated dataset of 69,970 compounds was retrieved. It is important to avoid inhibition of the cytochrome P450 (CYP) detoxification enzymes in order to minimize drug–drug interactions. Further screening using Lipinski’s filter reduced the total to 11,968 compounds. These were shortlisted, and the predicted activity values of the screened and refined compounds for inhibition of ABCB1 were estimated using our in-house Grind model [32]. Finally, after completion of the virtual screening pipeline (Figure 1), six potential test compounds with the highest predicted IC_50_ values were identified.

The lipophilicity and partition coefficients of the compounds were calculated (Table 2), and each of the six compounds were tested experimentally for potency and efficacy for inhibition of ABCB1 using our cellular drug transport assay.

### 2.4. Flow Cytometric Assay of Drug Efficacy and Potency: ABCB1 Inhibition Validated Using Tariquidar

We have previously integrated a single copy of pcDNA5/FRT-ABCB1-12His into the genome of Flp-In 293 cells to generate the Flp-In-ABCB1 cell line [33]. To prevent possible drug–drug interactions during the transport assay [34], the Flp-In-ABCB1 cells were cultured in the absence of hygromycin, which is normally used to select for the continued expression of the hygromycin resistance gene carried on the pcDNA5/FRT plasmid. This resulted in a loss of ABCB1 expression in a fraction of the Flp-In-ABCB1 cell line (Figure 2), which we exploit as an internal negative control. We also used the non-toxic cell-permeant dye calcein-AM as a transport substrate of ABCB1 [35,36].

The gating strategy is shown in Figure 2A,B shows that expression of ABCB1 correlates with reduced calcein accumulation by the cells. The concentration of calcein-AM was titrated until the ABCB1-positive cells began to accumulate calcein. Figure 2C shows that ABCB1-positive cells began to turn green when incubated with 250 nM calcein-AM. This concentration was used in all further live cell transport assays. To test whether the assay could be applied to drug inhibition studies, we incubated the Flp-In-ABCB1 cells with calcein-AM in the presence of the third-generation inhibitor tariquidar. It is clear from Figure 2D that 1 nM tariquidar fully inhibited ABCB1, as the Flp-In-ABCB1 population accumulated calcein to the same level as the parental Flp-In cells.

Importantly, Figure 2D also shows that, in the absence of tariquidar, approximately one third of the cells in the Flp-In-ABCB1 sample (population G4) accumulated calcein to the same level as the Flp-In cells. These are the cells that no longer express ABCB1 (population G3 in Figure 2B) due either to deletion of the plasmid from the genome or, perhaps more likely, silencing of the locus by chromatin condensation. These cells survive in the absence of hygromycin selection, but they are useful as they provide an internal negative control that can be easily identified in the single channel histogram at the lowest level of inhibitor. Transport activity of the ABCB1 was, thus, calculated using the fold difference in median drug accumulation by this ABCB1 non-expressing population G4 at the lowest concentration of inhibitor versus the median drug accumulation across all cells (population G5), as described in Equation (1).
(1)$$\frac{Median\;drug\;accumulation\;in\;ABCB1\;non-expressing\;cells\;\left(G4\right)}{Median\;drug\;accumulation\;in\;the\;whole\;population\;\left(G5\right)}$$

Figure 2 indicates that tariquidar is efficacious and can fully inhibit ABCB1. To test whether we can also use this assay to measure potency, we incubated Flp-In-ABCB1 with 250 nM calcein-AM and increasing concentrations of tariquidar. The histograms in Figure 3A show the representative raw data of the calcein content of the Flp-In-ABCB1 cells in the presence of increasing concentrations of tariquidar, while Figure 3B shows a non-linear regression analysis of the biological triplicate data. The IC_50_ for tariquidar inhibition of ABCB1 transport activity of 114 picomolar shows that this assay is extremely sensitive. This is likely due to the presence of a single copy of the ABCB1 cDNA in the Flp-In-ABCB1 cell line (thus ensuring that ABCB1 is unlikely to be over-expressed) and measurement of the partitioning of transport substrate directly (rather than, for example, modulation of cytotoxicity to a co-administered drug).

### 2.5. Four of the Six Test Compounds Fully Inhibit ABCB1 for the Transport of Calcein-AM but with Different Potencies

To determine the half-maximal inhibition concentrations for each of the test compounds ‘A’, ‘B’, ‘C’, ‘D’, ‘E’, and ‘F’, the Flp-In-ABCB1 cells were incubated with 250 nM calcein-AM and a range of concentrations of the test compounds. The accumulation of calcein in the cells was determined using flow cytometry, and the activity of ABCB1 calculated as above. It was apparent from preliminary data that compounds ‘B’ and ‘C’ had no effect on ABCB1 (at least not up to a concentration of 100 nM). We did not pursue compounds ‘B’ and ‘C’ further. Exemplar raw data for the remaining test compounds along with non-linear regression analysis of triplicate biological repeats are shown in Figure 4. The histograms in Figure 4A–D indicate an increase in calcein accumulation in Flp-In-ABCB1 cells with the increasing concentration of test compounds ‘A’, ‘D’, ‘E’, and ‘F’, respectively, while Figure 4E shows a non-linear regression analysis for all four that is comparable to tariquidar for efficacy; the mean bottom plateau value for tariquidar (0.78) and test compounds ‘A’ (1.0), ‘D’ (0.58), ‘E’ (0.93), and ‘F’ (0.36) are statistically equivalent (as the measure of ABCB1 activity is a ratio of the calcein content of ABCB1 non-expressing compared to the whole population, full inhibition is achieved at 1.0). The IC_50_ of compound ‘A’ was measured to be 1.4 nM with a 95% CI (0.4, 6.0), which is in close agreement with the predicted IC_50_ value of 6 nM calculated using our in-house Grind model [32], but significantly different (*p* = 0.0011) to tariquidar, which is 12-fold more potent in our assay system (tariquidar IC_50_ = 114 pM 95% CI (54 to 228 pM)). The IC_50_ of test compound ‘D’ was measured to be 23.6 nM with a 95% CI (11.3, 54.5), which is just outside the predicted IC_50_ value of 7 nM calculated using the Grind model. The IC_50_ of test compound ‘D’ differs significantly (*p* =< 0.0001) from tariquidar and also (*p* =< 0.0005) test compound ‘A’; thus, test compound ‘D’ appears to be 200-fold less potent than tariquidar and 17-fold less potent than test compound ‘A’. The IC_50_ of test compounds ‘E’ and ‘F’ were measured as 14.6 nM, 95% CI (5.6, 44.9), and 4.8 nM, 95% CI (1.7, 12.9), respectively, which are in close agreement with the respective predicted IC_50_ values of 7 nM and 2 nM calculated using the Grind model. The potency of test compounds ‘E’ and ‘F’ are significantly different (*p* =< 0.0001) to tariquidar, which is, respectively, 128-fold and 42-fold more potent in our assay system. The IC_50_ of test compound ‘E’ is also significantly different to test compound ‘A’ (*p* = 0.0138). However, the potency of test compound ‘F’ is statistically indistinguishable from test compound ‘A’. The *p* values for pairwise comparisons of potency (IC_50_ values) and efficacy (the bottom of the curves) are given in Table 3.

### 2.6. Compounds A and D Potentiate the Cytotoxicity of Taxol in ABCB1-Expressing Cells

The four compounds are potent inhibitors of calcein-AM transport, but can they inhibit the efflux of chemotherapeutic drugs. To address this question, we first substituted a fluorescent derivative of the anticancer drug taxol for calcein-AM in a transport assay. Figure 5A shows that all four compounds are able to inhibit the efflux of OREGON-GREEN taxol bisacetate (OG-taxol) by ABCB1. We next tested whether the drugs would sensitise the Flp-In-ABCB1 cells to the clinically relevant form of taxol. The Flp-In-ABCB1 cells and their parental Flp-In cells were challenged over a three-day period with a range of concentrations of taxol in the presence and absence of test compounds at 100 nM concentration. Figure 5B shows that compounds ‘A’ and ‘D’ resensitised the Flp-In-ABCB1 cells to taxol, which, in the presence of inhibitor, are as sensitive to taxol as the Flp-In parental cells. Furthermore, compounds ‘A’ and ‘D’ had no effect on the growth of the Flp-In parental cells, suggesting little or no cytotoxicity at this concentration (Figure 5C). In contrast, compounds ‘E’ and ‘F’ at 100 nM concentration did kill cells over the three-day period.

## 3. Discussion

In the present investigation, we identified new test compounds to target human ABCB1 with the potential to overcome MDR using a combined pharmacoinformatic approach, supported by experimental validation of compound efficacy and potency. We built a ligand-based pharmacophore model using a dataset of 98 inhibitors [24,25,26,27,28] to highlight the common structural features of selectivity for ABCB1 inhibition. To identify the top-scoring conformations of the compounds, the selective dataset was docked in the binding pocket of human-ABCB1 modelled on the cryo-EM coordinates (6QEX). This represents an advance from most previous studies, which lacked structural information on human-ABCB1, relying instead on homology models of human ABC transporters [16,32,37,38,39,40], which may have limited veracity of the structural features of previous pharmacophore models.

To build our pharmacophore model, we used an active analogue approach to select template Q37, similar to the strategy used by Noreen et al. to build a pharmacophore model based on a smaller dataset of quinoline derivatives [41]. Our ligand-based pharmacophore model postulated one hydrogen bond acceptor (HBA), three hydrophobic groups (Hyd1, Hyd2, and Hyd3), and one aromatic (Aro) feature for the selection of highly potent inhibitors of ABCB1. Previous efforts to develop pharmacophore models for ABCB1 inhibition consistently identified the presence of hydrophobic groups and hydrogen-bond acceptors as essential features [13,26,42]. However, the distance matrices varied slightly from our results. Zhang et al. [42] developed a pharmacophore model on the basis of 16 compounds, and reported three hydrophobic groups and one hydrogen bond acceptor as the crucial pharmacophoric features. Similarly, Ilza et al. [26] delineated three hydrophobic groups and a hydrogen bond acceptor as the important features on the basis of aligning three active compounds, while Kaczor et al. [13] used a dataset of 17 arylideneimidazolone derivatives including five active inhibitors and twelve inactive compounds to identify the importance of an aromatic group and hydrogen bond acceptor. Our pharmacophore model included all of the aforementioned features because we used a diverse dataset of 98 compounds that were described as selective for ABCB1 and that we were able to dock into a molecular model based on structure data for the human transporter. Our pharmacophore model scored highly for both the accuracy value (0.89) and MCC value (0.72) [29,30,43].

With the statistically significant pharmacophore model for inhibition of ABCB1 developed, we applied a previously reported pipeline of virtual screening for the identification of novel potential hits [30,41,43]. The potential hits were docked into the binding pocket of human ABCB1, and six compounds (‘A’, B’, ‘C’, ‘D’, ‘E’, and ‘F’) with the lowest predicted IC_50_ values [32] were identified. These test compounds were then validated for efficacy and potency to inhibit the efflux of calcein-AM by human ABCB1 in live cells and compared to the reference inhibitor tariquidar. In a confirmation of our in silico approach, four of the six test compounds showed efficacy to fully inhibit of ABCB1. Compounds ‘A’ and ‘F’ had the highest potency in the low nanomolar range, with ‘D’ and ‘E’ 10-fold lower. All four were also able to inhibit the efflux of OG-taxol from cells. Compounds ‘A’ and ‘D’ were also able to inhibit ABCB1 in cell culture and resensitise the cells to taxol. In these latter 72 h experiments, compounds ‘E’ and ‘F’ were found to be cytotoxic at 100 nM.

The lipophilic efficiency (LipE) of the compounds was also calculated. LipE can be a useful predictor of high inhibitory potency and an important tool in the test optimization process (Table 2). Previous studies have also shown that compounds with LipE values greater than 5 in combination with the CLogP values between 2 and 3 are optimal for oral bioavailability [40,41,43]. Kaczor et al. [13] and Jabeen et al. [40] estimated the lipophilicity of their respective arylideneimidazolone and benzophenone derivatives, but none achieved the standard threshold values for LipE and CLogP [13,40]. In the present study, the characteristics of two of the four test compounds (compounds ‘D’ and ‘F’) fulfilled the criteria for oral bioavailability, with ‘A’ and ‘E’ only marginally outside the optimal range for CLogP.

## 4. Materials and Methods

### 4.1. Computational

#### 4.1.1. Dataset Collection

An inhibitor dataset selective for ABCB1 of 98 compounds with known inhibitory potency (IC_50_) values in the range 0.05 to 113 µM was obtained from the reported literature [24,25,26,27,28]. Briefly, the ABCB1 dataset contains the derivatives of amide, ester, alkyl amine, polymethoxy, hydroxyl-N-phenyl, benzamide, ether, quinazolinone, and pyrrolopyrimidines (Appendix A, dataset of selective compounds). The three-dimensional structures of all compounds were built and cleaned using the Accelrys Draw 4.1 software (Symyx Technologies, Inc., Santa Clare, CA, USA. To obtain the most stable conformations of the compounds, the Molecular Operating Environment (MOE 2019.01; Montreal, QC, Canada) [43] was used to protonate each structure by titrating all the atoms, with 80% solvent at pH 7.0. This was followed by the energy minimization in MOE using the Merck Molecular Force Field 94x (MMFF94x; Merck and Co., Rahway, NJ, USA).

#### 4.1.2. Molecular Docking Simulation

To obtain the best docked conformations, the cryo-electron microscopic (cryo-EM) structure of human-ABCB1 (6QEX) with 3.6 Å resolution [17] was used for the molecular docking simulation of the dataset of 98 compounds using the software package GOLD (5.6.1) [44]. Rotamers of the side chains of ABCB1 were sampled to remove any bias during the pose generation step. The whole transmembrane region of 6QEX was considered as the likely binding pocket with X = 170.3330, Y = 165.0030, and Z = 169.2680 as coordinates. From published mutagenesis data, the presence of amino acids F728, Y310, Y307, I306, F303, W232, F343, L339, Q725, F983, F336, Q990, Q347, E875, S344, Q946, A871, L65, M949, M68, Y953, and M69 was considered important [17]. The binding coordinates define the cavity described by the transmembrane domains of ABCB1. To perform the docking simulation, a GOLD score was used, and 100 poses for each compound of the dataset were generated. Finally, the top-scoring conformation of each compound was selected for pharmacophore modelling.

### 4.2. Template Selection and Pharmacophore Modelling

For pharmacophore modelling, the active analogue approach [45,46] was used to select the most active template from the selective dataset of 98 docked compounds. Among the 75 highly active compounds of the ABCB1 dataset, the best docked conformation of the highly potent compound Q37 (Appendix A, dataset of selective compounds) with a reported IC_50_ value of 0.05 µM was selected as a template for the generation of a ligand-based pharmacophore model. Pharmacophore modelling was performed using the pharmacophore query editor, implemented in MOE [43] through the selection of ligand annotation points in the query editor of MOE.

### 4.3. Pharmacophore Model Validation

A threshold of <10 μM was defined for the active compounds (75 compounds) and ≥10 µM for the inactive compounds (23 compounds). For the assessment of model quality and the difference between predicted and actual values, Matthew’s correlation coefficient (MCC) (Equation (2)) and accuracy (Equation (3)) were used. MCC is the measure of the effectiveness of the binary classification ranging from −1 (no correlation) to 1 (full correlation) [2], and represents the functions of four variables: True Positives, TP; True Negatives, TN; False Positives, FP; and False Negatives, FN [1], as shown in Equation (1).
MCC = TP ∗ TN − FP ∗ FN/√ (TP +FP) ∗ (TP + FN) ∗ (TN + FP) ∗ (TN + FN)(2)
Accuracy= TP + TN/TP + FN + TN + FP(3)

In Equations (2) and (3):

TP = True Positives.

TN =True Negatives.

FP = False Positives.

FN = False Negatives.

The pharmacophore model was further used for virtual screening.

### 4.4. Pharmacophore-Based Virtual Screening

The pharmacophore model was used for the virtual screening of the ChemBridge database [23] to identify new chemical entities exhibiting binding potential against ABCB1. Briefly, the ChemBridge database was preprocessed to remove any inconsistencies and duplicates. In addition, different filters including molecular weight < 200 Da and >500 Da and logP were used for further data refinement. The compounds in the preprocessed database were energy-minimized using the MMFF94 force field, after which 50 conformations of each compound were generated stochastically in MOE [43].

The conformational database of the shortlisted compounds was further filtered against the in-house hERG model [30] to exclude inhibitors of human erythroblast transformation-specific transcription factor, and also filtered against five cytochromes, namely CYP 1A2, 2C9, 2C19, 2D6, and 3A4, using the online chemical modelling environment (OCHEM) [31]. The extracted compounds (CYP non-inhibitors and hERG non-blockers) were further screened against our ABCB1 pharmacophore model to obtain the putative active compounds. Predicted IC_50_ values were estimated using our in-house Grind model [32] to identify the four compounds likely to have the highest potency for experimental evaluation.

### 4.5. In Vitro Transport Inhibition Studies

Flp-In-293 and Flp-In-ABCB1-12His cells [33] were grown as monolayers in Dulbecco’s modified Eagle medium (DMEM; ThermoFisher Sci, Waltham, MA, USA) supplemented with 10% foetal bovine serum (FBS; ThermoFisher Sci, Waltham, MA, USA). The cell lines were incubated in a humidified incubator at a temperature of 37 °C in an atmosphere of 5% CO_2_. For the drug transport assay, cells (3 × 10^5^ per well) were seeded in a 24-well, flat-bottomed plate (Sarstedt AG and Co., Nümbrecht, Germany) in 500 μL growth medium and incubated for 24 h, as above. The medium was aspirated, and the cells were washed with the transport buffer (phenol-red free DMEM, with 1% FBS). Next, the cells were incubated with calcein-AM (Invitrogen, Waltham, MA, USA) at 250 nM or OG-taxol (Invitrogen, Waltham, MA, USA) at 0.4 μM in 200 μL transport buffer for 30 min at 37 °C in the presence or absence of inhibitor (Tariquidar was purchased from MedChemExpress (Monmouth Junction, NJ, USA) and the test inhibitors were from Chembridge Corporation (San Diego, CA, USA); Table 2). All test compounds, tariquidar, and calcein-AM were solubilized in DMSO and further diluted in transport buffer for the transport assay. The concentration of DMSO in the final solution never exceeded 1.1%, at which concentration it has no apparent effect on the cells or the function of ABCB1. The medium was then aspirated, and cells were washed with ice-cold transport buffer and harvested with 25 μL trypsin (TrypLE express enzyme; ThermoFisher Sci, Waltham, MA, USA). The cells were recovered in a flow cytometry tube in 300 μL ice-cold transport buffer for flow cytometry using a BD LSR II flow cytometer (Becton Dickinson, Franklin Lakes, NJ, USA).

In order to correlate ABCB1 expression with calcein-AM efflux activity, a slightly different order was used, as described previously [47]. Briefly, the cells were first detached from the well and incubated with saturating levels of primary antibody 4E3 (0.5 μg, Abcam, Cambridge, UK) in 100 μL transport buffer on ice for 30 min. The cells were recovered by gentle centrifugation and resuspended in 100 μL transport buffer with calcein-AM (250 nM), to which was added 2.5 μg secondary antibody, R-phycoerythrin-conjugated polyclonal goat anti-mouse (Dako, Santa Clara, CA, USA). After 30 min at 37 °C, the cells were pelleted, washed, and transferred to tubes, as above, for flow cytometry. After hydrolysis of the acetomethoxy in the cytosol, calcein was recorded in the green channel, while the phycoerythrin-conjugated secondary antibody was recorded in the red channel. Emission data were collected from 10,000 cells of normal size and granularity using Cell Quest^TM^ (Becton Dickinson, Franklin Lakes, NJ, USA). Data were analysed in Flow Jo software (Version 10; Becton Dickinson, Franklin Lakes, NJ, USA) followed by non-linear regression analysis in Graphpad PRISM (MacOS Version 9.4.1; GraphPad Software, San Diego, CA, USA) to determine the IC_50_ values for each compound and compare efficacy and potency by applying an extra-sum-of-squares F test. Triplicate biological repeats were completed for all compounds unless otherwise stated.

### 4.6. Taxol Challenge Assay

Flp-In-293 and Flp-In-ABCB1-12His cells (1 × 10^4^) were seeded into a 96-well dish in 100 μL of medium, as above. Taxol (Cambridge Bioscience, Cambridge, UK) was added to a final concentration ranging from 0 nM to 10 μM in the presence or absence of inhibitors at 100 nM. The cells were cultured for a further 72 h, after which the medium was aspirated, and the cells were detached with 30 μL TrypLE Express (ThermoFisher Sci, Waltham, MA, USA) before quenching with 75 μL transport buffer and transfer to flow cytometry tubes. Cells of normal size and granularity, gated on the zero-drug condition and then applied to all samples, were counted in an ACEA NovoCyte flow cytometer (Agilent Technologies, Santa Clara, CA, USA). Cell number data were analysed in Graphpad PRISM (MacOS Version 9.4.1; GraphPad Software, San Diego, CA, USA). Curve fitting and statistical analysis was performed as above.

### 4.7. Lipophilic Efficiency of Potential Hits

Lipophilic efficiency (LipE) is a combination of potency and lipophilicity [40] and estimates drug likeness. LipE of the potential hits (‘A’, ‘D’, ‘E’, and ‘F’) were calculated using PIC_50_ (calculated from IC_50_ values of compounds) and CLogP values (ChemBridge database, San Diego, CA, USA), using Equation (4).
LipE = PIC_50_ − CLogP(4)

In Equation (4), PIC_50_ and CLogP represent the biological potency and lipophilicity of the compounds, respectively.

## 5. Conclusions

Remarkably, four out of the six new compounds identified using our pharamacophore model were found to inhibit ABCB1 with high potency, providing proof-of-principle of our computational approach and also the utility of the cryo-EM structure data for this purpose. Preliminary characterization of the four compounds has identified compound ‘A’ as the most promising lead to target human ABCB1 with proven potential to overcome MDR in cell culture. Compound ‘D’ was also efficacious but with a slightly lower potency in transport assays. Compounds ‘E’ and ‘F’ may yet prove useful for laboratory studies. We expect that these compounds compete for the transport binding pocket, but further work will be required to fully characterize this mechanism and to realise their potential in the clinic. At present, compound ‘A’ is the most promising, and in Figure 6, we show two docking solutions of the compound in the human ABCB1 model based on the MOE and GOLD software. Figure 6A shows the best solution for a single molecule of ‘A’ in which the compound overlaps with the binding pocket for taxol, while Figure 6B shows the solution for two molecules, which dock closer to the cytosolic face of the transmembrane domains. Further work will be needed to determine the stoichiometry of inhibition and the veracity of these in silico predictions, but the structural data now available suggest that two bound molecules may be necessary to inhibit ABCB1.

## Data Availability

Raw flow cytometry data are available on request from Prof. Linton, and computational datasets are available on request from Ishrat Jabeen.

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
