# Peer review of "Identification and Empiric Evaluation of New Inhibitors of the Multidrug Transporter P-Glycoprotein (ABCB1)"

_ijms, 2023, doi:10.3390/ijms24065298_

Round 1
Reviewer 1 Report
The authors describe identification of inhibitors of ABCB1 based on an in-silico pharmacophore model and by screening the Chembridge database. Although the approach and the experimental plan are very good, the conclusion about the inhibitory potency of the tested compounds is based on only one functional assay. However, the identified compounds could actually be inhibiting the efflux of calcein-AM by behaving as competitive substrates.
Major concerns:
- While the authors have demonstrated that compounds A-F inhibit Calcein-AM efflux, this only indicates that the compounds interact with ABCB1. But the inhibition could actually be a result of the competition between calcein-AM and the test compound. To determine whether compounds A-F act as substrates or non-competitive inhibitors of P-gp, the authors need to test their effect on the ATPase activity of ABCB1, and whether the transporter can confer resistance to these compounds in cultured cells by a cytotoxicity assay.
- The authors demonstrate the optimal pose for the molecular docking of drug A with P-gp (PDB 6QEX) to add further evidence to the pharmacophore. Regardless, they should discuss the fact that non-competitive inhibitors of ABCB1, such as tariquidar, encequidar, elacridar, and zosuquidar, have been demonstrated to bind in pairs at two separate sites in human ABCB1 cryo-EM structures. The authors need to address this discrepancy—why they observe binding of only one molecule of drug A, whereas two molecules of known inhibitors bind to P-gp. How many poses of compound A were analyzed?
- It is not clear whether the IC50 values for the 98 compounds that were first identified as active in the literature were determined experimentally or computationally.
- The method for the IC50 prediction was indicated as in-house Grind model, which is not described in the Materials and Methods section. Please provide the basis and detail the settings for it.
- For easier reading of the article, the authors should add the number of compounds that result from each step of the protocol while screening for ABCB1 inhibitors, and especially clarify the final number.
- Please provide an explanation for why you only tested 4 of the "X" compounds obtained after using the Lipinski's filter. What was the estimated inhibitory potency difference between compound #4 or (F compound) and the following compound?
- Please include the structure of tariquidar in Table 1 for comparison.
- The label for the x axis of "Calcein" leads the reader to believe that it is Calcein concentration.
Reviewer 2 Report
The paper by Cheema et al is a very nice study employing cheminformatics, molecular docking and flow cytometry to demonstrate the combination of approaches being able to yield novel inhibitors of ABCB1/P-gp. This is an important research question as 30 years of work by many labs has so far failed to identify ABCB1 inhibitors that are selective, clinically effective and safe.
I would like to see the following points addressed by the authors, most of which are relatively minor.
i) Lines 63-65 and supplementary table 1. Firstly could the authors define a little more clearly what they mean by “activity against ABCB1” : there are a plethora of ways to measure ABCB1 interaction and an IC50 value in one technique might not correlate with that obtained using another method. Secondly, I think the supplementary table is rather obscure. I am sure many of the compounds Q1 to Q98 have names, or have references that could be cited. I think this would be an incredible useful addition to Suppl Table 1 with this information included.
ii) Figure 1 is a bit hard to relate to the text; I would draw some boxes over the labels “F2 Aro” etc and replace those with the terminology you have adopted in the main text (i.e. Aro). I think the F4:Hyd is what you refer to as HYD2 etc..
iii) Lines 107-112 : I can’t see where hERG is defined (I assume it’s a potassium channel) and neither can I see your rationale for having a hERG filter. Please just add 1-2 lines about this please. The P450 filter is more obvious but might also benefit from a sentence of justification just in case.
iv) Line 125, 143 and 402: please check you are referring to Table 2 here not table 1
v) Lines 128-187 are rather laboured. The flow cytometry assay is well established for ABCB1 by Linton and others and probably doesn’t need to be explained in such depth. Can you shorten this? Can you also remove the neologism chromatinisation and replace by “silencing by chromatin condensation”
vi) Table 2: I think it is essential to add the ChemBridge “look-up” or reference for the 4 “hit” chemicals identified. In table 2 I would also reduce the number decimal points in the molecular mass to just 2 dp.
vii) Page 5, but throughout the m.s; can the authors comment on why the chemicals B and C are not mentioned anywhere. Your four hits are A, D, E and F which to the outside reader makes it look as though two other chemicals were identified (B and C) which were not inhibitors of ABCB1. If that is the case then I don’t think this weakens the paper by its inclusion. Or is it something more simple – e.g. unavailability of other compounds?
viii) Line 193: the number of degrees of freedom and R2 are irrelevant and can be omitted.
ix) Line 196 I would rephrase the section in brackets to “(i.e. ensuring that ABCB1 is unlikely to be over expressed)”
x) Figures 3-5: is the y-axis scale of 0-5 sensible? It seems to me that 0 is not possible and that a value of 1 for the bottom plateau might even be used to constrain the non-linear fitting.
xi) Lines 205-246 and Figures 4 and 5. I don’t see why this is presented separately as two figures and two blocks of text. The four compounds should be part of a single integrated narrative, with a single figure.
xii) Lines 220-226, 240-246 and Table 3. It is unclear here what statistical tests have been performed. Is this an ANOVA followed by pairwise testing, or is it a series of t-tests. I am fairly confident that ANOVA and post-tests would be the correct way to present the stats and so the authors should confirm this and revise if appropriate. In table 3 I would simple say “p = n.s.” where the p-value was over 0.05 so clean up the table and help the reader see the significant differences.
xiii) Lines 338-350; the docking section is ambiguous. The x,y,z coordinates defined – are they the geometric centre of the docking box, which then extends 30A in all three dimensions, giving you a 30x30x30A box? That’s how I read it, but please do clarify. The list of amino acids is unexplained. What purpose does this list serve in the docking? Was it used to define that pocket centre?
Reviewer 3 Report
Jabeen et al. developed a pharmacophore model based on the the known inhibitors of ABCB1, the pharmacophore model was utilized in the virtual screening to find novel potential drugs, and finally four possible compounds with low IC50 were identified. Many major concerns need to be clarified:
1- The data used to estimate the predictive ability of the pharmacophore model seems very unbalance. The number of positive ligands are much larger than the negative ligands. The unbalanced number would distort the MCC and accuracy of prediction.
2- The cutoff value was set to be IC50=10, why that number was used? If change the cutoff, how about the pharmacophore model prediction accuracy and the final results?
3- The major contribution of the paper is using a new pharmacophore model to do the virtual screening, however, the role of the model only decreased the canditates number from 400K to 150K. My question is, how about the real effects of the new model to screening results? If the step is abandoned, will the similar compound list be characterized? The authors should provide the comparison results.
4- why the authors use the cryo-em structure to do the docking? Even without crystal structure of human ABCB1, many high-resolution structures of homologous could be employed to build a better model for docking.
5- how many ligands have the inhibition ability in the top 10 compounds found by the scheme in the paper? Only the four compounds with highest scores?
6- The inhibitor A, D, E, F are distinct with each other, why these compounds can inhibitor the ABCB1? The authors should provide explanation.
Reviewer 4 Report
The manuscript “Identification and Empiric Evaluation of new inhibitors of the 2 multidrug transporter P-Glycoprotein (ABCB1)” concerns a very sensitive area of cancer research. However, the authors did not present their results in a readable form. There are many inaccuracies etc. The manuscript should be checked for English and re-written.
1. It is necessary to describe clearly in the "Abstract" section the goal and tasks of the study and a strategy for their implementation.
2. A "Conclusions" section must be included.
3. There are a lot of grammatic errors (absence of commas etc.) and mistakes like those on lines 418 (...calcein fluoresces in the green...) and 420 (...FL-2 channel (543-627nm) channel...).
In general, it is difficult to read the manuscript.
Round 2
Reviewer 1 Report
The revised manuscript is significantly improved. It still remains to be seen whether these compounds are substrates or inhibitors. Additional transport studies are required to address this question.
Reviewer 2 Report
thanks to the authors for making changes to this manuscript. I think it represents a really publishable study
Reviewer 3 Report
No more revisions are required
Reviewer 4 Report
All my concerns have been properly addressed.